# Multimodality Imaging of Sudden Cardiac Death and Acute Complications in Acute Coronary Syndrome

**DOI:** 10.3390/jcm11195663

**Published:** 2022-09-26

**Authors:** Giuseppe Muscogiuri, Andrea Igoren Guaricci, Nicola Soldato, Riccardo Cau, Luca Saba, Paola Siena, Maria Grazia Tarsitano, Elisa Giannetta, Davide Sala, Paolo Sganzerla, Marco Gatti, Riccardo Faletti, Alberto Senatieri, Gregorio Chierchia, Gianluca Pontone, Paolo Marra, Mark G. Rabbat, Sandro Sironi

**Affiliations:** 1Department of Radiology, Istituto Auxologico Italiano IRCCS, San Luca Hospital, Piazzale Brescia 20, 20149 Milan, Italy; 2School of Medicine, University of Milano-Bicocca, 20126 Milan, Italy; 3University Cardiology Unit, Department of Interdisciplinary Medicine, University of Bari, 70121 Bari, Italy; 4Department of Radiology, Azienda Ospedaliero Universitaria (A.O.U.), di Cagliari-Polo di Monserrato, 09124 Cagliari, Italy; 5Department of Medical and Surgical Science, University Magna Grecia, 88100 Catanzaro, Italy; 6Department of Experimental Medicine, Sapienza University of Rome, Viale Regina Elena, 324, 00161 Rome, Italy; 7Department of Cardiac, Neurological and Metabolic Sciences, San Luca Hospital, Istituto Auxologico Italiano IRCCS, 20149 Milan, Italy; 8Radiology Unit, Department of Surgical Sciences, University of Turin, 10124 Turin, Italy; 9Centro Cardiologico Monzino IRCCS, 20138 Milan, Italy; 10Department of Radiology, ASST Papa Giovanni XXIII, 24127 Bergamo, Italy; 11Division of Cardiology, Loyola University of Chicago, Chicago, IL 60611, USA; 12Edward Hines Jr. VA Hospital, Hines, IL 60141, USA

**Keywords:** ischemic cardiomyopathy, acute myocardial infarction, cardiac arrhythmias, myocardial edema, late gadolinium enhancement, ventricular thrombus

## Abstract

Sudden cardiac death (SCD) is a potentially fatal event usually caused by a cardiac arrhythmia, which is often the result of coronary artery disease (CAD). Up to 80% of patients suffering from SCD have concomitant CAD. Arrhythmic complications may occur in patients with acute coronary syndrome (ACS) before admission, during revascularization procedures, and in hospital intensive care monitoring. In addition, about 20% of patients who survive cardiac arrest develop a transmural myocardial infarction (MI). Prevention of ACS can be evaluated in selected patients using cardiac computed tomography angiography (CCTA), while diagnosis can be depicted using electrocardiography (ECG), and complications can be evaluated with cardiac magnetic resonance (CMR) and echocardiography. CCTA can evaluate plaque, burden of disease, stenosis, and adverse plaque characteristics, in patients with chest pain. ECG and echocardiography are the first-line tests for ACS and are affordable and useful for diagnosis. CMR can evaluate function and the presence of complications after ACS, such as development of ventricular thrombus and presence of myocardial tissue characterization abnormalities that can be the substrate of ventricular arrhythmias.

## 1. Introduction

Coronary artery disease (CAD) represents one of the most important causes of death, especially in low and middle-income countries, caused by acute coronary syndrome (ACS) or cardiac complication related to ACS [1,2,3]. Prevention of acute events or non-invasive evaluation after the development of ACS can be performed using a multimodality approach [4,5]. Theoretically, an approach based using different non-invasive technique in ACS could result in an economical advantage, reduction of invasive procedures, and identification of patients that can have poor outcome after ACS [6]. Electrocardiogram (ECG) and echocardiography still represent the first line techniques in the presence of ACS. Cardiac magnetic resonance (CMR) is mainly confined to the evaluation of acute myocardial infarction in the early stage [5]; in particular, the evaluation of myocardial tissue abnormalities such as the presence of myocardial edema, microvascular obstruction (MVO), and the presence of left ventricular thrombus, can play a key role in terms of prognostic stratification in the early stage [5]. In the acute setting, coronary computed tomography angiography (CCTA) can be extremely important to rule out CAD [7], identify presence of high-risk coronary plaque [8], or severe stenosis, that may lead to acute myocardial infarction [9]. Furthermore, the application of CCTA can be extremely important for the identification of differential diagnosis causing chest pain and increased troponin, such as acute myocarditis [10].

The growing application of artificial intelligence (AI) in cardiac imaging is rapidly changing the diagnostic and therapeutic approach of patients with ischemic cardiomyopathy [11]. Potentially, the impact of AI will rapidly modify the management of patients with suspected ACS providing information regarding stenosis [12] and risk stratification [13].

In this review, we will demonstrate the application of non-invasive, multimodality imaging for evaluation of CAD, and possible sequelae that can lead to sudden cardiac death (SCD).

## 2. ECG Findings

ECG is an indispensable tool that help physicians to find potentially fatal situations after ACS occurrence. ECG is useful to identify cardiac arrhythmias. Indeed, the predominant causes of SCD are represented by tachyarrhythmias, such as ventricular fibrillation (VF) and sustained ventricular tachycardia (VT), or bradyarrhythmia-asystole events [14,15,16,17]. Ventricular arrhythmias (VA) are temporally distributed in two phases: early acute phase (from incident event up to 72 h), when ischemic dynamic changes and reperfusion occur; and intermediate-chronic phase (weeks-to-months later), when remodeling occurs.

Premature ventricular contractions (PVC) are common in the early phase of ACS. Both primary VF and VPC (particularly R-over-T phenomenon) occur during the early phase of ST elevation myocardial infarction (STEMI), when electrical instability is predominant. Although R-over-T beats can trigger VF, the sensitivity and specificity of these electrocardiographic findings during monitoring is too low to identify patients at increased risk for VT [18]. The pathophysiologic mechanism of fatal VA is based on the concept that an acute destabilizing event acts on a susceptible myocardial substrate. In this scenario, a SCA causes a myocardial scar, which generates a biochemical, electrolyte, and mechanical dysfunction, and creates the substrate for multiple arrhythmic re-entrant pathways [19].

Primary percutaneous coronary intervention (PCI), combined with the use of beta-blockers, have reduced the incidence of VT or VF during the first 48 h after a myocardial infarction [20,21,22]. Nevertheless, many patients present recurrent VA within the first 48 h and their response to pharmacological antiarrhythmic agents is variable [20]. Approximately 10% of post-MI survivors remain at high risk of dying in the first months or years following hospital discharge (mortality >25% at 2 years) [23,24]. Patients who develop sustained VF or VT after 48 h after their index MI have a predominantly higher rate of all-cause mortality [20]. However, the relationship between early (within 48 h of the index MI) VF/VT and mortality remains controversial [20]. The risk of recurrent cardiac arrest is usually in the range of 10–20%. Patients who experienced recurrent arrhythmias have a mortality rate of 50% [25].

On the other hand, ischemic lesions may cause a conduction block at any level of the atrioventricular or intraventricular system [26]. First-degree and second-degree type one atrioventricular (AV) blocks do not appear to affect survival and they are more often associated with occlusion of the right coronary artery and ischemia of the AV node [27]. Type II second-degree AV block is generally transient after an inferior-posterior infarction and manifests itself with a narrow/junctional QRS escape rhythm. However, after infarction of the anterior-wall, the block usually located below the His-bundle could evolve into a complete block [28,29]. On ECG, an unstable escape rhythm with wide QRS complexes can occur with ventricular rate <40 bpm and ventricular asystole can suddenly occur [30]. However, SCD may also occur after mechanical complications of ACS [31,32]. As mentioned before, twelve-lead EKG is more useful to find arrhythmic complications with respect to mechanical complications of MI. On the other hand, ECG together with clinical and echocardiographic findings (see below, echocardiography section), may be useful to identify post-myocardial infarction acute mechanical complications that may lead to SCD [33,34]. For instance, a ventricular septal defect may lead to the signs of evolving myocardial infarction, Q wave in the respective coronary affected territory and ventricular arrythmias. Clinically, the patient may present with a new murmur or signs of acute heart failure and circulatory collapse.

A free wall rupture, a life-threatening condition, typically occurs after ineffective reperfusion therapy or delayed presentation of MI [35,36]. ECG may show a new ST-segment elevation, because outflow of blood irritates the pericardium. Patients may present with new episodes of chest pain, nausea, hemodynamic instability, or collapse.

Lastly, ECG may detect electrolyte imbalances, a condition typically observed in ACS patients, which represents a pro-arrhythmic burden [37,38,39]. For instance, hyperkalemia may be suspected in patients who present with increased voltage of the T-waves, with pointed, narrow and symmetrical morphology, widening of the QRS-duration (right bundle branch-block-like), reduction of the amplitude of the P-wave, and ST-segment elevation in the right precordial leads [40]. Similarly, hypokalemia is also common in this setting of patients, and it can cause VA [41]. It may cause reduction of the amplitude of the T wave until its disappearance, QT interval prolongation, atrioventricular blocks, and ST-segment depression [42].

The role of ECG in patient with ACS can be definitely extremely important for the assessment of complications due to AMI and identifications of patients that could potentially develop arrhythmogenic SCD.

A representative case is shown in Figure 1.

## 3. Echocardiography

Echocardiography is the first imaging tool used both in suspected or established ACS [43]. This is due both to its feasibility as a bedside diagnostic technique and to the relatively recent application to assess aspects of myocardial mechanics and advanced systolic and diastolic function [44,45,46,47]. The first aim of the echocardiographic examination in cases of ACS is to assess the presence and extension of kinesis abnormalities, which represent markers of ischemia (hypo-dys-akinesia) [44,48,49,50]. In addition, echocardiography helps physicians to assess overall ventricular volumes and left ventricle ejection fraction (LVEF), possible extension of ischemia to the right ventricle, subsequent pericardial disease, and possible mechanical complications [51,52]. In the ACS diagnostic pathway, echocardiography and secondary imaging exams, such as CCTA or CMR, could identify other potentially deadly pathologies as differential causes of symptoms: aortic dissection; right-sided overload from acute pulmonary embolism; hypertrophic cardiomyopathy; aortic stenosis; and pericardial pathology [53].

In the era of primary coronary reperfusion therapy for ACS, both mortality and incidence of mechanical complications (0.27%), such as intraventricular septal rupture, free wall rupture with or without tamponade, or acute mitral regurgitation (MR) have decreased [54,55,56,57,58]. Nevertheless, mechanical complications can lead to mortality and significant morbidity, especially in older patients, so they require early diagnosis and therapeutic management. Patients with mechanical complications tend to be older, female, have a history of heart failure, chronic kidney disease, and are often presenting with their first ACS [58,59,60]. In this scenario, prompt use of transthoracic echocardiography is essential [57,61,62,63].

Several studies showed that free wall rupture (FWR) is the most frequent mechanical complication post-MI presenting as a mild pericardial effusion, chest pain and nausea, and rapid development of cardiac tamponade, or as SCD due to electromechanical dissociation [64]. Echocardiography is useful to confirm the diagnosis and promptly guide surgical repair. Nevertheless, in-hospital mortality for FWR, even if treated surgically, still remain > 35% [65]. Although uncommon, aneurysms and pseudoaneurysms (PAN), which derive from an incomplete rupture of the LV free wall, may occur after ACS and have a high mortality rate up to 20% [66]. Aneurysms are located in the left ventricle apex in 80% of cases, while PANs are typically located is the anterior or lateral wall of the LV [67]. Echocardiographic features are the following: a cavity which communicates with the LV chamber through a large (aneurysm) or tight (PANs) neck; a thrombus into either aneurysms or PANs may be found; doppler and color flow imaging may show bidirectional blood flow via the site of rupture [68]. Although echocardiography is useful to promptly find aneurysms or PANs, sometimes-detected characteristics are less typical. In this case, cardiac magnetic resonance (CMR) may be useful [69].

The main echocardiographic findings of pericardial tamponade are represented by pericardial effusion, inferior vena cava (IVC) dilatation, diastolic right ventricular collapse, and systolic right atrial collapse [70]. IVC plethora consists of dilatation of the IVC > 20 mm without respiratory collapse (<50% of the basal diameter) and it is a very sensitive sign of cardiac tamponade (92%). Atrial collapse is usually observed before ventricular collapse during progressive cardiac tamponade. Right atrium collapse is usually observed in early systole, when intracavity pressure is lower. Furthermore, a prolonged duration of either atrial collapse (more than one-third of the cardiac cycle) and RV collapse have been described as highly sensitive and specific sign of clinical cardiac tamponade [71,72,73]. Ventricular septal defects (VSD) have an incidence of 0.3%, mainly due to early reperfusion therapy. Many studies show that VSD is generally caused by multivessel coronary disease [74,75] and generally occurs 3–5 days after an MI. It is characterized by a high mortality rate (85% of patients within 2 days) and evolution into cardiac failure and cardiogenic shock if not treated effectively [58,59]. VSD are more frequently located in the anterior than inferior/lateral wall (66% versus 34%). Although uncommon, inferior VSDs are more complex with multiple, irregular, and/or variable interventricular connections and they are associated with a higher mortality rate and SCD, probably due to contemporary right ventricle dysfunction [75]. Posterior VSDs are often associated with mitral regurgitation secondary to ischemic tethering. The rupture of the septum is shown as a partial opening during diastole and closure in systole. A left-to-right shunt via the VSD is usually found using color-doppler imaging. Patients with inferior/posterior VSD, early onset of symptoms and signs of right ventricular failure represent a subgroup at higher risk of SCD (Figure 2) [76,77,78].

Acute ischemic MR is a rare complication (0.05%), which typically occurs 2–7 days after AMI [61,79,80]. It is generally caused by the rupture of a papillary muscle or chorda tendineae [81]. Acute MR generates a massive blood flow which is ejected into the left atrium during ventricular systole. This causes a reduction of the ejection fraction and consequently a cardiocirculatory collapse. In addition, massive regurgitation via the incompetent valve leads to a volume and pressure overload in the left atrium, reflecting in higher pulmonary pressure and congestion [82]. Echocardiography is mandatory to assess the presence and severity of MR, especially in cases of cardiogenic shock. The direction of the regurgitant jet on color flow doppler may help physician to identify the etiology of the MR. In papillary muscle rupture, echocardiography can show a mobile mass prolapsing into left atrium during ventricular systole. However, only a leaflet prolapse may be found [83].

Prediction of ventricular arrhythmias after MI is challenging. Echocardiographic evaluation of the global systolic function assessed by LVEF still remains the primary approach to stratify patients with ACS at high risk of SCD [49]. SCD is most frequently caused by ventricular arrhythmias resulting from electrical and mechanical changes in the affected myocardium and may be prevented by an implantable cardioverter-defibrillator (ICD) [84]. It is well established that patients with LVEF < 35% after ACS present an improvement in overall survival by receiving an ICD, due to a reduction in arrhythmic fatal events [85,86]. Similarly, patients who receive biventricular pacing appear to have a lower rate of mortality due to tachyarrhythmias [84,87,88]. Several studies demonstrate that LV hypertrophy with eccentric remodeling and dilatation represent risk factors for SCD in ACS patients [88,89,90]. However, many patients suffering from SCD after AMI have a LVEF > 35%. This reflects a poor sensitivity of LVEF as a unique mechanical risk-stratifying parameter for this subgroup of patients [91]. In this scenario, myocardial strain represents a novel and accurate tool able to quantify global and regional myocardial function and the timing of contraction. Particularly, several studies have found that global longitudinal strain (GLS) is more accurate than LVEF in assessing LV function post-MI and in predicting mortality and ventricular arrhythmias [92,93]. GLS is a more accurate marker to detect subtle changes in myocardial function, which may be important in arrhythmic risk stratification [94,95]. Furthermore, a novel strain echocardiography parameter is mechanical dispersion, which is defined as the time to peak negative strain from the 16 LV segments and represents a measure of myocardial contraction heterogeneity [96,97]. Mechanical dispersion appears to be a new tool in the risk assessment of SCD because it predicts arrhythmias in patients after myocardial infarction independently by GLS and traditional parameters such as EF, volumes, and diastolic dysfunction [92].

Echocardiography still represents the first technique useful for the evaluation of AMI complications. In particular, it can be extremely helpful for the identification of patients that can develop structural complications and fatal arrhythmias with ACS.

## 4. Cardiac Magnetic Resonance

CMR can play a key role in patients with acute myocardial infarction but also with chronic ischemic cardiomyopathy [4,5,98,99,100,101]. In particular, the additional value of CMR over echocardiography is represented by the possibility to identify reversible or irreversible damaged myocardial areas, the latter of which could potentially be a trigger of arrhythmias [102,103]. The assessment of myocardial damage using CMR in ischemic cardiomyopathy can be differentiated considering the acute or chronic setting [4,104]. In the acute setting, it is possible to evaluate the presence of myocardial edema, MVO and late gadolinium enhancement (LGE) [5,105]; in patients with chronic ischemic cardiomyopathy, LGE can be useful for evaluation of viability as well as the presence of heterogeneous LGE, which is associated with increased incidence of arrhythmias [5].

Two cases of AMI evaluated with CMR are shown in Figure 3 and Figure 4.

## 5. Myocardial Edema and Area at Risk

In the past, the evaluation of myocardial edema in clinical practice has been performed using T2 black blood sequences [106]. When acquiring T2 black blood (T2BB) sequences, myocardial edema is hyperintense, normal myocardium has normal signal intensity, while blood pool shows black signal intensity [5]. The acquisition of T2BB is routinely performed in clinical practice using T2 short-tau inversion recovery (T2 STIR) and T2 spectral adiabatic inversion recovery (T2 SPAIR). The longtime of acquisition is one of the common disadvantages of both T2 STIR and T2 SPAIR; furthermore, T2 STIR can show artifacts due to misalignment of three radiofrequency pulses; T2 SPAIR can show artifacts related to presence of B_0_ inhomogeneities [107].

In order to overcome the limitations due to T2 BB images, the approach with T1 and T2 mapping has been developed [108]. Native T1 mapping is sensitive to increased extracellular volume; however, it is not specific for edema [109], conversely, T2 mapping is very sensitive and specific for presence of edema [110].

Evaluation of edema in acute myocardial infarction can be performed using qualitative or semiquantitative approaches [5] and in the acute setting allows the ability to identify edema and area at risk (AAR); the latter represents the mismatch between the edematous myocardium and myocardial area of LGE [5] and potentially the myocardium that can be salvaged by timely revascularization [5].

Comparing the T2 BB versus the T1 and T2 mapping approach, Bulluck et al., showed that the approach using mapping allows a minor variability in terms of inter and intrareader agreement for the evaluation of AAR [111]. Although AAR can be extremely important in the acute setting, it is extremely important to consider the “bimodal wave” of myocardial edema development in patients with acute myocardial infarction [112]. Indeed, the development of edema follows a dynamic process beginning with the development due to reperfusion, followed by edema associated with inflammation and remodeling [112]. Furthermore, it is important to consider that development of myocardial edema can be variable considering the myocardial territory supplied, the coronary configuration and the culprit lesion [113].

Zorzi et al., evaluated the impact of edema depicted on CMR in a mixed population comprising sudden cardiac arrest due to ischemic and non-ischemic cardiomyopathy. The authors found that myocardial edema was associated with a significantly higher survival free from ICD interventions (long rank: 0.04) and ICD shock (long rank: 0.03) [114]. Therefore, the authors hypothesize that myocardial edema represents only a transient arrhythmogenic substrate [114]. Similar results were observed in a smaller cohort of patients by Zorzi et al.; the authors showed that in patients who experienced out-of-hospital cardiac arrest, the presence of edema was associated with a favorable outcome [115].

Therefore, it appears that the presence of edema depicted on CMR represents a transient arrhythmogenic substrate for development of arrhythmias.

## 6. Microvascular Obstruction

The development of MVO during acute myocardial infarction can be related to different factors, such as ischemic injury, reperfusion injury, distal embolization, and individual predisposition [116,117]. The presence of MVO in the acute setting can be depicted using perfusion, early gadolinium (EGE), or LGE sequences, after the administration of a gadolinium-based contrast agent, demonstrated as a myocardial dark area with ischemic distribution in the damaged myocardium [5]. The presence of MVO decreased over time and amount during perfusion is usually greater compared to LGE [118]. Using a non-contrast approach, it is possible to depict MVO in native T1 mapping showing decreased values compared to surrounding myocardial tissue [119].

Usually, MVO is associated with left ventricle remodeling, increased left ventricle volume, and impairment of left ventricle ejection fraction, that lead to poor outcome of patients with acute myocardial infarction [120,121]. Despite the evaluation that was observed with myocardial contrast echocardiography, it seems that in the early phase the presence of no reflow areas is not associated with an increased development of arrhythmias [122], while in a short follow-up the presence of MVO and expression of irreversibly damaged myocardium, can lead to development of arrhythmias [123].

The development of MVO causes the development of intramyocardial hemorrhage (IMH), the expression of blood cell deposition, vascular endothelial damage, and iron deposition [124].

The depiction of IMH can be evaluated using T2* sequences demonstrating iron deposition within the damaged myocardium [125]; while native T1 mapping can be used for the depiction of MVO in the infarcted myocardium, differentiating it from salvageable and remote myocardium that shows higher values of native T1 [119].

The intramyocardial deposition of iron is associated with an arrhythmogenic trigger [126,127].

Cokic et al., in an animal model, compared the ECG findings between two subgroups showing absence and presence of iron deposition after ACS [127]. Animals showing iron deposition demonstrated an elevated QT interval and QT corrected (*p* < 0.05) during the day, night and 24 h, compared to animals negative for iron deposition [127].

Mather et al., analyzed the impact of IMH demonstrating that patients with the presence of IMH showed a significantly (*p* = 0.04) longer filtered QRS (125 ms) compared to patients without IMH (109 ms) [126]. Furthermore, the authors showed that the size of MVO was not an independent predictor of prolonged filtered QRS [126].

These manuscripts demonstrate that arrhythmias can be developed following the iron deposition within myocardium.

## 7. Ventricular Thrombus

The development of thrombus after ACS is caused by the presence of Virchow’s triad after an acute event comprising myocardial wall motion abnormalities, subendocardial inflammation, and systemic hypercoagulability due to ACS [128].

The presence of ventricular thrombus can be identified as a hypointense area inside the ventricular cavity using EGE sequences, in particular EGE should be acquired within two minutes after administration of the contrast agent [129]. LGE can be useful for the assessment of left ventricular thrombus showing a specificity of 99% and a sensitivity of 88% [130]. An approach without contrast can be feasible using native T1 mapping, which shows lower values compared to patients with blood pool [131]. Despite the incidence of thrombus after acute myocardial infarction is more frequent in the anterior or apical myocardial infarction, in patients with ACS the incidence of left ventricular thrombus detected using CMR ranges between 6.3% and 12.2%, respectively [128,132]. In the era of thrombolytic therapy, the presence of left ventricular thrombus causing stroke is around 2–3%, while limited data is still available for patients treated with percutaneous coronary intervention [128]. The thrombus that can lead to the development of emboli are mainly represented by a protruding shape and/or cavitary motion [133]. A CMR protocol dedicated for the evaluation of LV thrombus should be mandatory for the evaluation of LV thrombus.

## 8. Late Gadolinium Enhancement

To date, late gadolinium enhancement sequences are the most studied in the field of magnetic resonance, and its prognostic power has been demonstrated in numerous contexts of the cardiovascular field [134,135,136]. The evaluation of LGE using CMR in the acute setting could be important in terms of risk stratification of acute events, but also as sequelae of myocardial infarction [137,138]. The evaluation of LGE in acute myocardial infarction is performed 10–15 min after the administration of a gadolinium-based contrast agent [139,140,141]. The wash-out of gadolinium in the infarcted myocardial areas is decreased, therefore it is possible to obtain images with remote myocardium with a null signal, bright blood pool, and hyperintense signal intensity of the infarcted myocardium showing an ischemic pattern [4]. Black-blood LGE (BB-LGE) is another approach developed for the identification of the infarcted area [99,101,142]. Using BB-LGE it is possible to obtain hyperintense signal intensity of infarcted myocardium, black blood pool, and grey signal intensity of remote myocardium [101].

The main advantage of BB-LGE over the standard sequences used for the evaluation of LGE is represented by the possibility to better evaluate subendocardial infarction and infarcted papillary muscles [99,101].

In terms of arrhythmogenic stratification, the assessment of LGE in patients with ACS should be carefully evaluated considering that the amount of LGE can decrease over the time [5].

The presence of myocardial infarction heterogeneity and the development of ventricular arrhythmias have been explored by Robbers et al. [138]. The authors demonstrated that a large proportion of penumbra within the myocardial enhanced area can increase the risk of VT (*p* = 0.003); the latter was associated with an increased prevalence of sudden death [138]. Furthermore, the presence of ventricular fibrillation, prior PCI, and the proportional amount of penumbra within the enhanced area in a multivariate analysis are independently associated to VT [138].

During CMR, it is important to evaluate the presence of infarcted papillary muscles, which can trigger an arrhythmia [143,144]. Although the study was performed using echocardiography, the authors found that patients with infarcted papillary muscles showed a greater QRS (150 ± 15 ms) compared to patients with fascicular arrhythmias (127 ± 11 ms) (*p* = 0.001). Echocardiographic data were confirmed using CMR by Bogun et al. [137] who showed that heterogenous LGE was more frequent in arrhythmogenic papillary muscles compared with papillary muscles not involved in ventricular arrhythmias (*p* = 0.01) [137].

## 9. Cardiac Computed Tomography Angiography

According to the ESC and AHA guidelines, CCTA is considered one of the best non-invasive techniques to rule out the presence of CAD [145,146,147,148,149,150,151]. Despite the low positive predictive value in terms of ischemic coronary lesions [7], CCTA represents a unique tool for the noninvasive evaluation of plaque characteristics. The association between specific plaque features and adverse plaque characteristics has been linked with worse outcomes (Figure 5) [8,152,153]. Furthermore, CCTA it is a useful tool for the evaluation of anomalous origin or course of coronaries that can be associated with a worse outcome [154].

Tissue characterization is another interesting tool that can be extremely important for assessment of patients with chest pain and increased troponin (Figure 6) [155].

In summary, CCTA can be easily identify patients with characteristic of coronary plaque and stenosis or myocardial tissue abnormalities that can lead to development of SCD.

## 10. Coronary Stenosis and Plaque

The assessment of coronary plaque using CCTA may be indicative of patients that can develop sudden cardiac death [156,157,158,159].

Min et al., demonstrated that patients with stenosis < 50% have a higher survival rate (survival 99.7%) compared to patients with left main disease ≥ 50% (survival 85%) [160]. Beyond stenosis, Min et al., demonstrated that the location of disease was important in terms of survival [160]. The presence of significant proximal stenosis was associated with a worse outcome if the stenosis was evident in the left anterior descending artery [160].

Beyond the location of coronary stenosis, CCTA can play a key role in the non-invasive evaluation of plaque burden and adverse plaque characteristics that has been associated with worse outcomes [8].

Chang et al., demonstrated that more than 65% of patients developing acute coronary syndrome have non-obstructive CAD and 52% of cases have high risk plaque (HRP) [158]. In particular, the adjusted hazard ratio was 1.010 for percentage of diameter stenosis, while 1.593 for plaque showing necrotic core [158].

The impact of plaque characteristic has also been evaluated in the SCOT-HEART trial, showing that patients with adverse plaque showed death or myocardial infarction three times more frequent (*p* < 0.01 and HR: 3.01) and two times more frequent in patients with obstructive coronary artery disease (*p* = 0.02 and HR: 1.99) [159]. The combination of obstructive CAD and adverse remodeling demonstrated a 10-fold increase of death or myocardial infarction compared to patients with normal coronary arteries (*p* < 0.01 and HR: 11.50) [159].

Considering the impact of plaque characteristics and coronary artery stenosis in terms of prognosis, the reporting system of CCTA has been modified, taking into account that both metrics can be useful in terms of diagnosis and prognosis [161,162,163]; especially considering that patients after CCTA, beyond invasive coronary angiography, could be potentially treated with aggressive medical therapy [161].

## 11. CCTA in Emergency

CCTA in an emergency can [164] play a key role to rule out CAD in select cases [155], especially in patients with acute chest pain, increase of troponin and inconclusive diagnosis (discrepancy with marker, symptoms, ECG and echocardiography) [10].

The ROMICAT trial described the impact of CCTA in patients with acute chest pain [165]. Hoffmann et al., demonstrated that in patients with acute chest pain, CCTA was able to rule out 50% of patients with low to intermediate likelihood of ACS with a negative predictive value of 100% [165]. The ROMICAT II trial highlighted the importance of plaque characteristics in patients who underwent CCTA in an emergency setting [164]. Patients with ACS showed HRP more frequently (*p*: 0.006 with an OR: 8.9) [164].

Another useful approach of CCTA can be represented by tissue characterization in the acute setting [10].

Esposito et al., demonstrated that the application of CCTA in the emergency department can provide information regarding myocarditis or non-ischemic cardiomyopathies [155]. A study with a larger cohort of patients showed that using CCTA was possible to overcome the concept of “triple rule out” entering in the new era of “quadruple rule out” improving diagnosis [10].

## 12. Application of Artificial Intelligence to Non-Invasive Multimodality Imaging

The application of artificial intelligence (AI) in non-invasive cardiovascular imaging is rapidly growing [11]. In particular, the application of artificial intelligence can modify the approach to CAD in clinical practice speeding up the time of acquisition, reporting, and building models for risk stratification [12,100,166,167,168].

Focusing on echocardiography, the manuscript of Narang et al., demonstrated the possibility to evaluate left ventricle and left atrium volume in 35 ± 17 s [169]. Furthermore, algorithms trained for the depiction of regional wall motion abnormalities can provide results similar to experienced sonographer (AUC 0.99 for deep learning model vs. 0.98 for sonographers) [170].

In terms of CMR, the application of AI ranges from image quality to image analysis [100,171,172].

In particular, Moccia et al., demonstrated that it was possible to segment the myocardial scar using a fully convolutional neural network with a sensitivity and Dice score of 88.07% and 71.25%, compared to manual segmentation [172].

The application of AI in CCTA aimed to provide information regarding the stenosis, plaque characteristics, association with ischemia, and risk stratification [12,167,173].

Evaluation of coronary stenosis has been described by several authors founding a good agreement for the depiction of coronary artery stenosis [12,174,175]. Furthermore, AI algorithms can be used for risk stratification as demonstrated by Motwani et al. [176]. In particular, the authors demonstrated that combining the data of machine learning (ML) with clinical data was possible to predict better the risk of all-cause mortality, compared to clinical score in a follow up of 5 years [176]. The impact of ML has been demonstrated also by Van Rosendael et al. [177]. The authors demonstrated that an algorithm based on ML is able to stratify better patients compared to current CCTA integrated risk score [177].

In summary, the application of AI can definitely speed up the time of reporting of imaging modalities and it could potentially provide data regarding prognosis.

## 13. Future Perspective

A multimodality approach in the application of cardiovascular disease plays a key role for the prevention of sudden cardiac death due to coronary artery disease.

Therefore, it is not surprising that a multimodality imaging approach combined with artificial intelligence algorithms may change the management of patients with CAD and reduce the risk of fatal arrhythmias.

## Figures and Tables

**Figure 1 jcm-11-05663-f001:**
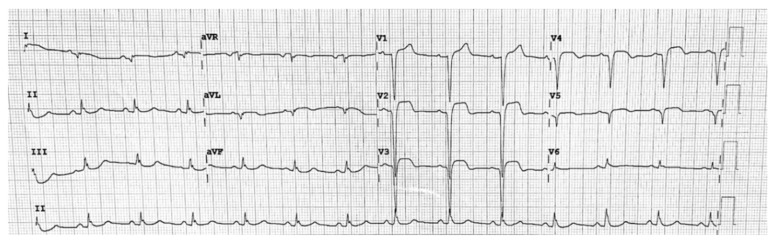
Twelve-lead EKG from a 60 years old patient five days after admission in our referral hospital for antero-lateral ST-segment elevation myocardial infarction (STEMI). Patient underwent primary revascularization of left descending coronary artery. Five days after reperfusion, patient experienced hypotension and new chest pain episode. EKG showed a new ST-segment elevation in the anterior precordial leads. Prompt bedside echocardiography showed an acute mild-to-moderate pericardial tamponade. The origin site of this tamponade was not observed.

**Figure 2 jcm-11-05663-f002:**
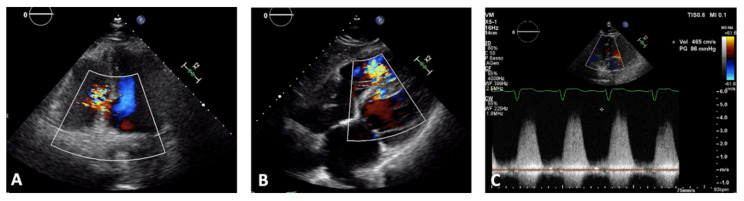
Ventricular septal defect (VSD) in a patient diagnosed with subacute myocardial infarction 3 days after reperfusion of right coronary artery. (**A**) Apical four-chamber view shows presence of a posterior VSD. Color Doppler mode demonstrated a high-velocity flow through the neck of the VSD, which corresponds to left-to-right shunt. (**B**) Subcostal view confirmed presence of a serpiginous posterior VSD with left-to-right shunt. (**C**) Continuous-wave Doppler imaging showed presence of a high velocity shunt, although underestimated because of non-completely parallel positioning of doppler-marker as respect to flow direction.

**Figure 3 jcm-11-05663-f003:**
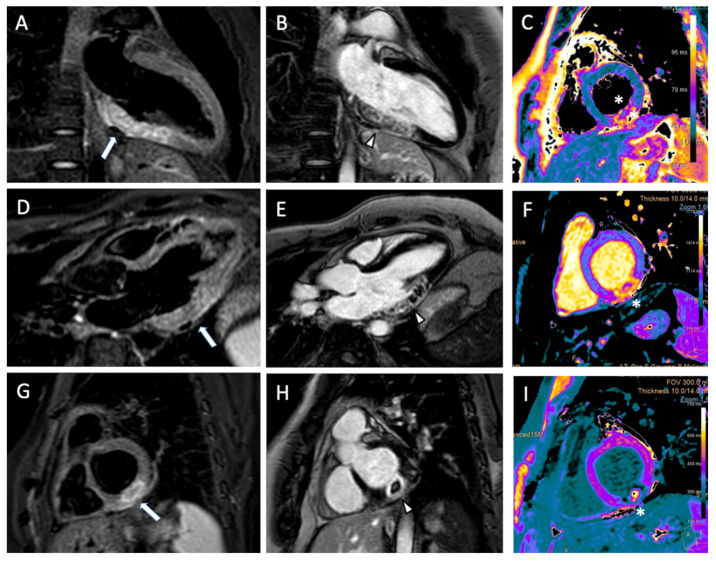
(**A**) 41 year old male patient underwent cardiac magnetic resonance for acute myocardial infarction. T2 black blood images show myocardial edema on inferolateral wall in two chamber (arrow, (**A**)), three chamber (arrow, (**D**)), and short axis (arrow, (**G**)). The myocardial infarcted area with presence of microvascular obstruction was observed on late gadolinium enhancement in two chamber (arrowhead, (**B**)), three chamber (arrowhead, (**E**)), and short axis (arrowhead, (**H**)). Microvascular obstruction was observed on short axis T2 mapping (asterisk, (**C**)), native T1 mapping (asterisk, (**F**) e post-contrast native T1 (asterisk, (**I**)).

**Figure 4 jcm-11-05663-f004:**
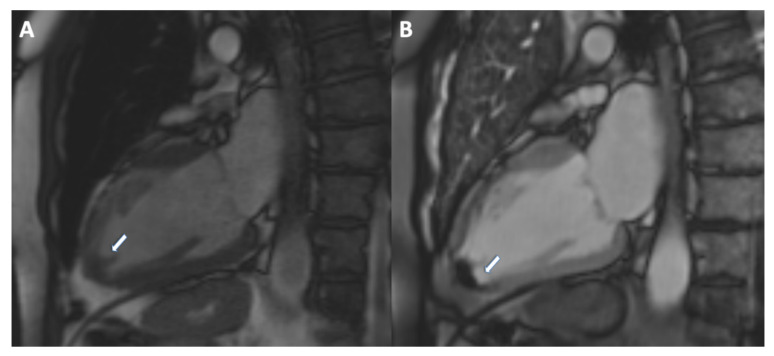
(**A**) 72 year old male patient with previous AMI on LAD territory. The patient was admitted to hospital due to chest pain. Images show possibility of a left ventricular thrombus on two chamber cine image ((**B**), arrow). Finding was confirmed on two chamber early gadolinium enhancement ((**B**), arrow).

**Figure 5 jcm-11-05663-f005:**
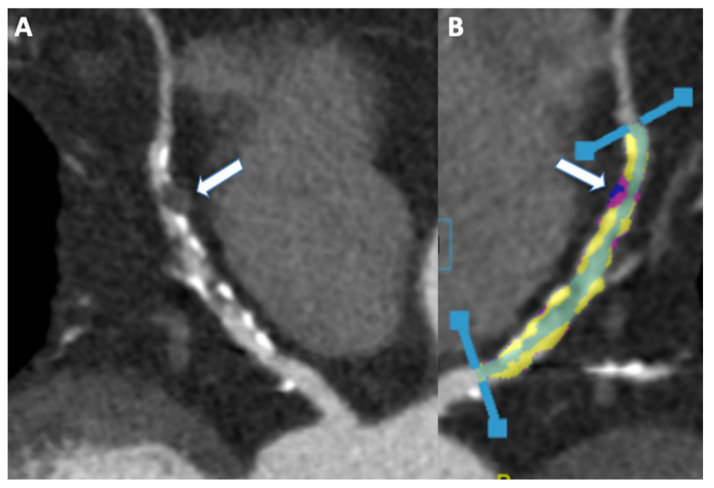
(**A**) 55 year old male patient showing diffuse coronary artery disease on left anterior descending artery (panel (**A**)). The predominant plaques were calcified, a fibro-fatty plaque with positive remodeling was observed on mid-LAD (arrow, (**A**)). Plaque analysis confirm huge coronary calcification (yellow, (**B**)), while fibrofatty and remodeled plaque (arrow, (**B**)) show fibrofatty plaque (purple) associates with necrotic core (blue).

**Figure 6 jcm-11-05663-f006:**
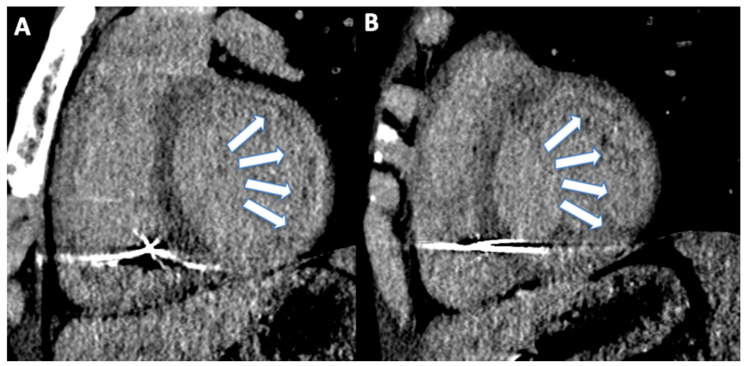
(**A**) 23 year old male patient with history of cardiac arrest. After the implantation of implantable cardioverter defibrillator, the patient acquired cardiac computed tomography angiography for the assessment of late iodine enhancement. Late iodine enhancement reconstructed on short axis showed intramyocardial late iodine enhancement on anterior, lateral, and inferior wall of basal (arrows, (**A**)) and mid-ventricular (arrows, (**B**)) left ventricle.

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
