# Peer review of "Multimodality Imaging of Sudden Cardiac Death and Acute Complications in Acute Coronary Syndrome"

_jcm, 2022, doi:10.3390/jcm11195663_

Round 1

Reviewer 1 Report

Muscogiuri et al wrote this review to overview the different imaging method to evaluate the heart and vascular function. It is an informative article for a budding scientist. In general, please work on formatting the article much better to make it more readable. Below are some examples: 

1.     Please see how to reword the title of the review to reflect the content of the review better, as of now, it is not very self-evident, the topic of the review is not limited to sudden cardiac death.

2.     Many paragraphs of the article are organized in the way of either sentences or a summarization of a reference article, e.g. line 250, page 12 etc. please reorganize so that at least one paragraph would summarize one major aspects.

3.     There are some data that came from the authors, are the data published? If yes, please reference them, if not, please indicate that.

4.     For “cardiac magnetic resonance”, the term is “cardiac magnetic resonance imaging”, with magnetic resonance imaging abbreviated as MRI, please make corresponding changes if you agree, otherwise, please explain.

5.     Please read through the article carefully to make sure there is no misspellings, e.g. “although” in line 326.

Author Response

Muscogiuri et al wrote this review to overview the different imaging method to evaluate the heart and vascular function. It is an informative article for a budding scientist. In general, please work on formatting the article much better to make it more readable. Below are some examples: 

Q1: Please see how to reword the title of the review to reflect the content of the review better, as of now, it is not very self-evident, the topic of the review is not limited to sudden cardiac death.

A1: We agree with reviewer that the title is misleading. According the suggestion of reviewer we modified the title as follow: “ Non Invasive Multimodality imaging of Sudden Cardiac Death  and Acute Complications in Acute Coronary Syndrome “

Q2: Many paragraphs of the article are organized in the way of either sentences or a summarization of a reference article, e.g. line 250, page 12 etc. please reorganize so that at least one paragraph would summarize one major aspects.

A2: We agree with reviewer and we thank the reviewer for this fuitful comment.

Following the suggestion we add a paragraph to each techcnique, excluding CMR where we resumed the role of each technique and the impact that can have in terms of identification for SCD.

Q3: There are some data that came from the authors, are the data published? If yes, please reference them, if not, please indicate that.

A3: We agree with reviewer that some paragraph need to be justified with a reference. Therefore we modified the manuscript accordingly especially in the session regarding ECG and Echocardiography.

Q4: For “cardiac magnetic resonance”, the term is “cardiac magnetic resonance imaging”, with magnetic resonance imaging abbreviated as MRI, please make corresponding changes if you agree, otherwise, please explain.

A4: We thank the the reviewer for the comment however we think that CMR is more useful as term for indication of cardiac magnetic resonance considering the wide use of CMR in literature. The term MRI would be too vast.

Q5: Please read through the article carefully to make sure there is no misspellings, e.g. “although” in line 326.

A5: We thank the reviewer for the comment. We evaluated the manuscript throughout the entire length and we modified it  accordingly.

Reviewer 2 Report

This review article is describing use of multimodal imaging for acute coronary syndrome with some extension to prognostication for sudden cardiac death. 

General explanations are more focused on imaging of ACS. The title is more focused on sudden cardiac death. The title should be modified to fit to the contents of the review better.

line 75-77, terminologies are mixed (ECG, Electrocardiogram, EKG) need to check all documents and use abbreviation.

line 353, full words for HRP should be included once. 

line 378. Before Future perspective, I suggest making separate section for application of AI and include examples. It was summarized with only a few sentences with future perspective without any details.

line 933, undergone -> underwent

Need to check all abbreviations again.

Author Response

This review article is describing use of multimodal imaging for acute coronary syndrome with some extension to prognostication for sudden cardiac death. 

Q1: General explanations are more focused on imaging of ACS. The title is more focused on sudden cardiac death. The title should be modified to fit to the contents of the review better.

A1: We agree with reviewer for the comment. Following the suggestion we modified the manuscript as follow: “Multimodality imaging of Sudden Cardiac Death  and Acute Complications in Acute Coronary Syndrome”

Q2: line 75-77, terminologies are mixed (ECG, Electrocardiogram, EKG) need to check all documents and use abbreviation.

A2: We Thank the reviewer for the comment and we apologize for the mistake. We modified the manuscript accordingly following the suggestion of the reviewer.

Q3: line 353, full words for HRP should be included once. 

A3: We thank the reviewer. We modified the manuscript accordingly.

Q4: line 378. Before Future perspective, I suggest making separate section for application of AI and include examples. It was summarized with only a few sentences with future perspective without any details.

A4: We thank the reviewer for the comment. The manuscript has been changed focusing on application of AI in ischemic cardiomyopathy. Considering the wide topic it has been summarize in only one paragraph.

Q5: line 933, undergone -> underwent

A5: We thank the reviewer for the comment. We modified the manuscript accordingly.

Q6: Need to check all abbreviations again.

A6: We thank the reviewer for the comment. We modified the manuscript accordingly.

Reviewer 3 Report

This review is not well focussed- just a compendium of  imaging techniques described for various patients with coronary disease.

There is no new information and one is left wondering what is the takehome message for clinicians. 

This manuscript lists various imaging modalities that may be used to evaluate patients with coronary disease.

It is entirely unclear what the patient population the authors are referring to is- those who present with SCD, those who present with MI, and are at risk for subsequent SCD?

.

The authors fail to mention the most important imaging approach to ACS- namely invasive coronary angiography to identify extent and severity of disease. Determination of plaque characteristics by advanced imaging modalities such as OCT, near infrared spectroscopy  are also not mentioned.

The authors concentrate on the utility of various non invasive adjunctive imaging techniques- but they are not placed into clinical context- when in the course of disease are these useful- do the authors believe, that MSCTA is an important modality in patients who present with ACS as opposed to invasive angiography?-

The authors should recast the manuscript to clearly state the patient population they are referring to, describe the various clinical time points when imaging techniques are of benefit, and describe how the findings from these studies  may influence outcome.

Author Response

This review is not well focussed- just a compendium of  imaging techniques described for various patients with coronary disease.

Q1: There is no new information and one is left wondering what is the takehome message for clinicians. 

 A1: We thank the reviewer for the fruitful comment. Following the suggestion of the this reviewer and previous reviewer we modified the paragraph for each technique enhancing the role for each non invasive technique.

Q2: This manuscript lists various imaging modalities that may be used to evaluate patients with coronary disease. It is entirely unclear what the patient population the authors are referring to is- those who present with SCD, those who present with MI, and are at risk for subsequent SCD?

A2: We understand the concerning of the reviewer regarding the population analyzed by the manuscript. The article is mainly focused on population with CAD that can develop SCD due to arrhythmias or complications of AMI. Furthermore the impact of each non invasive technique is analyzed  in order to prevent the develop AMI. In order to better clarify this concept we add the following sentence at the end of the introduction: In this review, we will demonstrate the application of non invasive multimodality imaging for evaluation of CAD and possible sequelae that can lead to sudden cardiac death (SCD).  

Q3: The authors fail to mention the most important imaging approach to ACS- namely invasive coronary angiography to identify extent and severity of disease. Determination of plaque characteristics by advanced imaging modalities such as OCT, near infrared spectroscopy  are also not mentioned.

A3: We agree with reviewer that this point can be fundamental, however considering the length and the focus of this manuscript we decided to not mention invasive coronary angiography. Furthermore we decided to change the title of the article as follow: Multimodality imaging of Sudden Cardiac Death  and Acute Complications in Acute Coronary Syndrome

Q4: The authors concentrate on the utility of various non invasive adjunctive imaging techniques- but they are not placed into clinical context- when in the course of disease are these useful- do the authors believe, that MSCTA is an important modality in patients who present with ACS as opposed to invasive angiography?

A4: We agree with reviewer that it is better to specify the patients where CCTA can be useful in Emergency. Following the suggestions we add the following sentence: CCTA in an emergency can play a key role to rule out CAD in select cases, especially in patients with acute chest pain, increase of troponin and inconclusive diagnosis (discrepancy with marker, symptoms, ECG and echocardiography).

Q5: The authors should recast the manuscript to clearly state the patient population they are referring to, describe the various clinical time points when imaging techniques are of benefit, and describe how the findings from these studies may influence outcome.

A5: We thank the reviewer for the comment. Following the suggestion of this reviewer and the advices of previous reviewer we modified accordingly adding:

  • A key message regarding the impact of each technique for depiction of SCD in patients with acute event.
  • We modified the title, highlighting that the review will focus on non-invasive imaging.
  • We added that, following the new CAD-RADS classification patients can be treated also with aggressive medical therapy.
  • We also included that AI can play a key role in terms of prognosis.

Round 2

Reviewer 3 Report

I do not think the auth9ors have adequately addressed my comments.

They have presented a wealth of information, but the review lacks focus  and does not give clinicians an adequate guide to the role and timing of imaging modalities

I would suggest the manuscript be recast to emphasize a specific clinical problem within CAD, and explain the role of imaging modalities for that specific problem.